# Graph Learning for Numeric Planning

**Dillon Z. Chen**[1,2]  **Sylvie Thiébaux**[1,2]
[1]LAAS-CNRS, University of Toulouse  [2]The Australian National University
{dillon.chen,sylvie.thiebaux}@laas.fr

## Abstract

Graph learning is naturally well suited for use in symbolic, object-centric planning due to its ability to exploit relational structures exhibited in planning domains and to take as input planning instances with arbitrary numbers of objects. Numeric planning is an extension of symbolic planning in which states may now also exhibit numeric variables. In this work, we propose data-efficient and interpretable machine learning models for learning to solve numeric planning tasks. This involves constructing a new graph kernel for graphs with both continuous and categorical attributes, as well as new optimisation methods for learning heuristic functions for numeric planning. Experiments show that our graph kernels are vastly more efficient and generalise better than graph neural networks for numeric planning, and also yield competitive coverage performance compared to domain-independent numeric planners. Code is available at `https://github.com/DillonZChen/goose`

## 1 Introduction

Planning requires long range reasoning over combinatorially large state spaces. Numeric planning is an extension of classical planning in which states have numeric variables and the underlying transition system is built from inequality conditions and assignments over arithmetic expressions of such variables. It was formalised in PDDL 2.1 [FL03] and is undecidable in the general case [Hel02] which makes it more difficult than classical planning which is PSPACE-complete [Byl94]. Numeric planning is a well-established problem in the symbolic AI community and exhibits significant research effort [CCFL13, IM17, SHTR20, KSP+22, KSB23, SKB23], but this expressivity result implies that building a general, scalable numeric planner is a challenging problem.

Learning for Planning (L4P) is a research direction which focuses on learning to solve problems from a specified domain in an automated supervised manner [TTTX20, STT20, FGT+22, KS21, SBG22, SBG23, MLTK23, CTT24a, SDS+24, RTG+24, APK24]. Planning tasks in L4P are assumed to exhibit a factored, symbolic representation, which allow us to generate training data in a matter of seconds from easy to solve tasks with a domain-independent planner. We can then learn domain knowledge in a supervised manner that scales planners to significantly larger planning tasks.

This is in contrast to Reinforcement Learning where agents do not require access to well-defined models but spend significant amounts of time exploring and learning from rewards [SB98]. Regardless, several works have showed that encoding or learning symbolic models for sequential decision making reasoning and embodied AI tasks [LCZ+21, ZYP+22, LSS+22, SCK+23, KVS+23, LPM23] provided better performance and transparency over end-to-end reinforcement learning methods. Furthermore, it was shown recently that classical ML methods are much better suited for L4P than deep learning methods for symbolic planning [CTT24b] as they (1) can generalise well from small training data, (2) are orders of magnitude more efficient to train and evaluate than deep learning methods, which is important in time sensitive tasks such as planning, and (3) have interpretable features to understand what is being learned.

In this paper we study whether this fact carries over to Learning for Numeric Planning (L4NP) [WT24] which now requires reasoning over logic and arithmetic. It is reasonable to think that because neural networks are function approximators, they may offer better reasoning capabilities over numbers than

38th Conference on Neural Information Processing Systems (NeurIPS 2024).

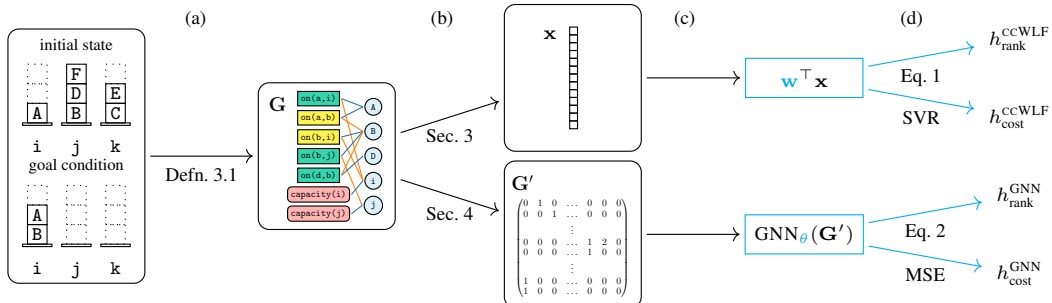

Figure 1: The GOOSE framework for learning heuristic functions for numeric planning. Cyan colours indicate components that are influenced by the training phase. (a) A numeric planning state and goal condition is encoded into a graph $\mathbf{G}$ via the $\nu$ILG representation defined in Defn. 3.1. (b) Graphs are either embedded into vectors $\mathbf{x}$ in Euclidean space with the CCWL kernel from Sec. 3 or transformed into a graph $\mathbf{G}'$ with a real-valued matrix representing node features as inputs into GNNs described in Sec. 4. (c) Features $\mathbf{x}$ are fed into a linear model, whereas transformed graphs $\mathbf{G}'$ are fed into GNNs. (d) Linear models are either trained by the ranking formulation in Eq. 1 or by Support Vector Regression (SVR) with a linear kernel. GNN models are either trained by the ranking formulation in Eq. 2 or by backpropagation minimising the mean squared error (MSE) loss.

just symbols alone. In this paper, we describe the GOOSE[1] framework with classical ML and deep learning configurations for learning heuristic or value functions for use with search in L4NP. Fig. 1 illustrates the GOOSE framework and we outline our contributions as follows.

- We introduce a new graph representation of numeric planning tasks for use with classical and deep graph learning methods, namely graph kernels and graph neural networks, respectively.
- We extend the WL kernel [SSVL+11] to handle graphs with both continuous and categorical attributes in a meaningful way which we call the CCWL kernel.
- We introduce new ranking formulations [GKL16, CEKP23, HTT+24] for learning heuristic or value functions with linear programs.

The structure of the remainder of the paper is as follows. In Sec. 2, we provide the necessary formalism and background for numeric planning, as well as relevant notation. In Sec. 3, we introduce a new graph encoding $\nu$ILG and CCWL kernel for generating features for numeric planning tasks. In Sec. 4, we introduce a deep learning architecture for L4NP using graph neural networks. In Sec. 5, we describe optimisation methods for L4NP, involving a new ranking formulation for learning heuristic functions. In Sec. 6, we describe our experimental setup and results. Related work is discussed in Sec. B in the appendix. We conclude the paper with final comments in Sec. 8.

## 2  Background

**Numeric Planning Task.** A numeric planning task can be viewed as a compact representation of a deterministic, goal-conditioned Markov Decision Process with the use of predicate logic and relational numeric variables. A majority of the remainder of this section formalises the necessary components of a numeric planning task we use in the paper.

A numeric planning task [FL03] is given by a tuple $\Pi = \langle X_p, X_n, A, s_0, G \rangle$ where $X_p$ is a finite set of propositional variables with domain $\{\top, \bot\}$ and $X_n$ is a finite set of numeric variables with domain $\mathbb{R}$. Let $X = X_p \cup X_n$ denote the set of state variables, where a state is a total assignment of values the propositional and numeric variables. The variables implicitly induce a possibly infinite set of states $S$, where $s_0$ is the initial state.

A propositional condition is a positive (resp. negative) literal $x = \top$ (resp. $\bot$) for some propositional variable $x \in X_p$, and a numeric condition has the form $\xi \unrhd 0$ where $\xi$ is an arithmetic expression over numeric variables and $\unrhd \in \{\geq, >, =\}$. We write $[x]^s$ (resp. $[\xi]^s$) for the value of a state variable $x$ (resp. expression $\xi$) in state $s$, and $V(\xi)$ for the set of numeric state variables in $\xi$. A state $s$ satisfies a set of conditions (i.e. a set of propositional and numeric conditions) if each condition in the set evaluates to true given the values of the state variables in $s$. The goal $G$ is a set of conditions and we write $G_p$ (resp. $G_n$) for the subset of propositional (resp. numeric) goal conditions.

---

[1]**G**raphs **O**ptimised f**O**r **S**earch **E**valuation

The set $A$ contains a finite number of actions, each consisting of preconditions and effects. Action preconditions $\text{pre}(a)$ are sets of conditions, and action effects assign Boolean values to propositional variables and assign the value of an arithmetic expression to numeric variables. An action $a$ is applicable in a state $s$ if $s$ satisfies $\text{pre}(a)$, in which case its successor $a(s)$ is the state where the effects $\text{eff}(a)$ are applied to the state variables in $s$. If $a$ is not applicable in $s$, we set $a(s) = s_\perp \notin S$. Each action $a$ has a cost $c(a)$ given by an arithmetic expression.

A plan for a numeric planning task is a sequence of actions $\pi = a_1, \ldots, a_n$ such that $s_i = a_i(s_{i-1}) \neq s_\perp$ for all $1 \le i \le n$ and $s_n$ satisfies $G$; we call $s_0, s_1, \ldots, s_n$ the plan trace of the plan. The plan length $|\pi|$ and the plan cost are the number of actions in the plan, and the sum of their cost, respectively. A plan is optimal if it has the minimum cost among all plans. A numeric planning task is solvable if there exists a plan for it, and is unsolvable otherwise. A state $s$ is a deadend if the task with the initial state replaced with $s$ is unsolvable. Satisficing planning refers to the problem of finding a plan if it exists, or proving that the problem is unsolvable. Optimal planning refers to the problem of finding an optimal plan if it exists, or proving that the problem is unsolvable.

**Lifted representation.** Numeric planning tasks can be compactly encoded in a lifted representation $\langle \mathcal{O}, \Sigma_p, \Sigma_f, \Sigma_a, \mathcal{A}, s_0, G \rangle$ whereby state variables are derived from a set of predicates, functions, and objects. Formally $\Sigma_p$ and $\Sigma_f$ are sets of predicate and function symbols, respectively. Each symbol $\sigma \in \Sigma_p \cup \Sigma_f$, has an arity $n_\sigma \in \mathbb{N} \cup \{0\}$ which depends on $\sigma$. Predicates and functions take the form $p(x_1, \ldots, x_{n_p})$ and $f(x_1, \ldots, x_{n_f})$, respectively, where the $x_i$s are their arguments. Given the set $\mathcal{O}$ of objects, the propositional and numeric variables are obtained by substituting objects for the arguments of the predicates and functions, resulting in the grounded form $p(o_1, \ldots, o_{n_p})$ and $f(o_1, \ldots, o_{n_f})$, respectively, where the $o_i$s are objects. Similarly, actions can be represented in a lifted form via a set $\Sigma_a$ of action symbols and a set $\mathcal{A}$ of action schemata mapping action symbols to their lifted precondition and effect definitions in terms of predicates and functions. Grounding the set of action schemata results in the set of actions $A$ of the planning task. Details are not needed to understand this paper. A domain is a set of numeric planning tasks sharing the same set of $\Sigma_p$, $\Sigma_f$, $\Sigma_a$, and $\mathcal{A}$, and may have constant objects, objects which are shared across all tasks in the domain.

**Example: Capacity Constrained Blocksworld.** To help digest some of the definitions of numeric planning, we provide an example with a planning domain we call *Capacity Constrained Blocksworld* (ccBlocksworld). It is an extension of the original Blocksworld domain in which state consists of towers of blocks and the objective is to stack and unstack blocks to achieve a goal configuration. It is also a special case of the Hydraulic Blocksworld domain for planning with state constraints, in which blocks are placed on top of pistons which rise or fall depending on the configurations of other pistons [HIR+18].

In ccBlocksworld, we have a maximum number of tower locations, and each tower has a base limited by the number of blocks it can hold. To model this domain in the lifted representation, we retain the predicate $\text{on}(x, y)$ from the original Blocksworld, which indicates block $x$ is on another block or base $y$. Next, we also introduce the function $\text{capacity}(z)$ which denotes the remaining number of blocks that are allowed to be placed on base $z$. The numeric variables instantiated from $\text{capacity}$ may increase or decrease depending on whether blocks are unstacked from the tower or stacked on top of it. Action schemata preconditions constrain whether a block can be placed on a tower with base that has reached its capacity limit or not. The leftmost figure in Fig. 2 illustrates an example of a ccBlocksworld problem with an initial state and goal condition. We refer to the Sec. A of the appendix for the complete state representation of the problem as well as its PDDL encoding.

**Heuristics and Greedy Best First Search.** State-of-the-art methods for both satisficing and optimal numeric planning [SHTR20, KSP+22, CT24] employ some variant of heuristic search. A heuristic function maps a state $s$ to $\mathbb{R} \cup \{\infty\}$ representing an estimate of the cost to reach the goal from the current state, where a value of $\infty$ estimates that $s$ is a deadend. The optimal heuristic $h^*$ maps a state to the cost of an optimal plan if it exists, and $\infty$ otherwise. The Greedy Best First Search (GBFS) algorithm consists of a priority queue initialised with the initial state as the only element, and a main loop which performs the following steps while the queue is non-empty: (1) pop a state $s$ with the lowest heuristic value and some tie-breaking criterion from the queue, (2) generate the successors of $s$ via all applicable actions, and (3) check if a successor $s'$ is a goal, in which case terminate with the plan traced back from $s'$, and otherwise add $s'$ to the queue if it has not been seen before. The algorithm determines a problem is unsolvable if the main loop completes, in which case the problem has finitely many states of which all have been seen.

**Graph and other notations.** Let $\mathbf{G} = \langle \mathbf{V}, \mathbf{E}, \mathbf{F}_{\mathrm{cat}}, \mathbf{F}_{\mathrm{con}}, \mathbf{L} \rangle$ denote a graph with nodes $\mathbf{V}$, undirected edges $\mathbf{E} \subseteq \binom{\mathbf{V}}{2}$, categorical node features $\mathbf{F}_{\mathrm{cat}} : \mathbf{V} \to \Sigma_{\mathrm{V}}$ where $\Sigma_{\mathrm{V}}$ is a finite set, continuous node features $\mathbf{F}_{\mathrm{con}} : \mathbf{V} \to \mathbb{R}^d$ with $d \in \mathbb{N}$, and edge labels (categorical edge features) $\mathbf{L} : \mathbf{E} \to \Sigma_{\mathrm{E}}$ where $\Sigma_{\mathrm{E}}$ is a finite set. The neighbourhood of a node $u \in \mathbf{V}$ in a graph with respect to an edge label $\iota$ is defined by $\mathbf{N}_\iota(u) = \{v \mid \exists e \in \mathbf{E}, \text{ s.t. } e = \langle u, v \rangle = \langle v, u \rangle \wedge \mathbf{L}(e) = \iota\}$. We use $\|$ to denote the concatenation operator for vectors, and $[\![N]\!]$ to denote $\{1, \ldots, N\}$.

# 3 Relational features for numeric planning

In this section, we describe an automatic method for generating embeddings for numeric planning tasks that may be used for any downstream inference model. The method is an extension of the feature generation method for classical planning [CTT24b] and consists of two main steps: (1) generating a graph representation of a planning task, and (2) running a variant of the WL-algorithm for generating features for the graph [SSVL$^+$11]. Extending the first step of the method is simple as it is easy to extend the graph encoding to capture numeric information of the task. This is done in Sec. 3.1 where we introduce the Numeric Instance Learning Graph ($\nu$ILG) representation for numeric planning tasks. The second step is more difficult as we require constructing a WL-algorithm variant that can handle both categorical and continuous nodes features in a meaningful way for numeric planning. This is where we introduce the CCWL algorithm in Sec. 3.2 that handle such node features. Thus, we can generate features for numeric planning tasks by first converting them into the $\nu$ILG representation, and then running the CCWL algorithm on them.

## 3.1 Graph encoding of numeric planning tasks

We begin by describing our graph encoding of a planning task, namely the Numeric Instance Learning Graph ($\nu$ILG). Similarly to the classical case, the graph representation does not encode the transition model of the planning task nor requires grounding all possible variables in the planning task. Thus, our encoding only requires a first-order representation of states, and therefore applies to problems whose transition model is unknown such as in model-free reinforcement learning.

We begin with a descriptive definition of the graph with an example Fig. 2 illustrating a subgraph of the $\nu$ILG representation of the example ccBlocksworld problem. In the figure, the nodes in the graph represent the objects (light blue), propositional variables true in the state (green), numeric variables (red), propositional goals (yellow) and numeric goals (not present in the example) of the problem. Blue (resp. orange) edges connect object nodes to goal and variable nodes where the object is instantiated in the first (resp. second) argument of the corresponding node variable or condition.

We provide the formal definition below. Let $X_p(s)$ denote the set of propositional variables that are true in $s$, $X_n(s)$ the set of numeric variables, and $X(s) = X_p(s) \cup X_n(s)$.

**Definition 3.1** (Numeric Instance Learning Graph). The *Numeric Instance Learning Graph ($\nu$ILG)* of a numeric planning task in the lifted representation $\Pi = \langle \mathcal{O}, \Sigma_p, \Sigma_f, \Sigma_a, \mathcal{A}, s_0, G \rangle$ is a graph $\mathbf{G} = \langle \mathbf{V}, \mathbf{E}, \mathbf{F}_{\mathrm{cat}}, \mathbf{F}_{\mathrm{con}}, \mathbf{L} \rangle$ with

- nodes $\mathbf{V} = \mathcal{O} \cup X(s_0) \cup G$, where we assume w.l.o.g. that $V(g) \subseteq X(s_0)$ for all $g \in G_n$,

- edges $\mathbf{E} = \bigcup_{p = \sigma(o_1, \ldots, o_{n_\sigma}) \in X(s_0) \cup G_p} \{\langle p, o_i \rangle \mid i \in [\![n_\sigma]\!]\} \cup \bigcup_{\xi \trianglerighteq 0 \in G_n} \{\langle \xi, v \rangle \mid v \in V(g)\}$,

- categorical node features $\mathbf{F}_{\mathrm{cat}} : \mathbf{V} \to \Sigma_{\mathrm{V}}$ with $\mathbf{F}_{\mathrm{cat}}(u) =$

$$
\begin{cases}
\text{OBJ}(u) & \text{if } u \in \mathcal{O} \\
\text{FUNC}(u) & \text{if } u \in X_n(s_0) \\
(\text{COMP}(u), \text{ACH}(u)) & \text{if } u \in G_n
\end{cases}
\begin{cases}
(\text{PRED}(u), \texttt{achieved\_propositional\_goal}) & \text{if } u \in X_p(s_0) \cap G_p \\
(\text{PRED}(u), \texttt{unachieved\_propositional\_goal}) & \text{if } u \in G_p \setminus X_p(s_0) \\
(\text{PRED}(u), \texttt{achieved\_propositional\_nongoal}) & \text{if } u \in X_p(s_0) \setminus G_p
\end{cases}
$$

  where $\text{OBJ}(u) = u$ if $u$ is a constant object and $\texttt{object}$ otherwise, $\text{PRED}(u)/\text{FUNC}(u)$ returns the predicate/function symbol from which a proposition/fluent was instantiated from, $\text{COMP}(u) \in \{\geq, >, =\}$ encodes the comparator type of the numeric goal condition $u$, and $\text{ACH}(u) \in \{\texttt{unachieved\_numeric\_goal}, \texttt{achieved\_numeric\_goal}\}$ encodes whether $s_0$ satisfies $u$,

- continuous node features $\mathbf{F}_{\mathrm{con}} : \mathbf{V} \to \mathbb{R}$ where $\mathbf{F}_{\mathrm{con}}(u) = [u]^{s_0}$ for nodes $u \in X_n(s_0)$, $\mathbf{F}_{\mathrm{con}}(u) = [\xi]^{s_0}$ for nodes $u = \xi \trianglerighteq 0 \in G_n$ with $[\xi]^{s_0} \ntrianglerighteq 0$, and $\mathbf{F}_{\mathrm{con}}(u) = 0$ otherwise, and

- edge labels $\mathbf{L} : \mathbf{E} \to \Sigma_{\mathrm{E}}$ where for edges of the form $e = \langle p, o_i \rangle$, we have $\mathbf{L}(e) = i$, and otherwise for edges $e = \langle \xi, v \rangle$, we have $\mathbf{L}(e) = 0$.

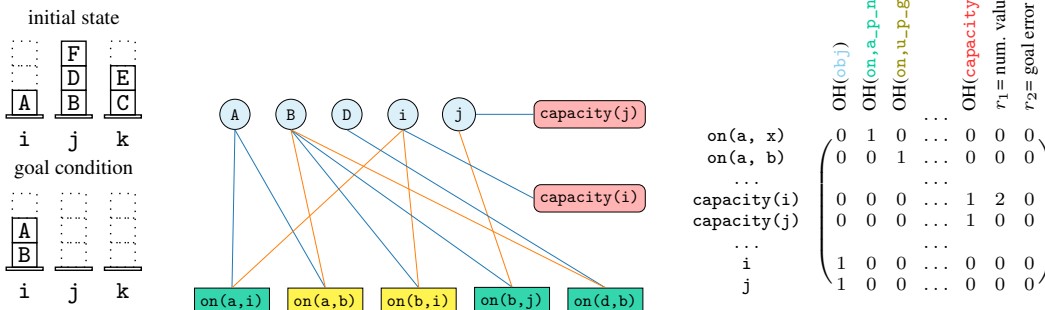

Figure 2: An example ccBlocksworld task where each base has capacity 3 (left), a subgraph of its ILG representation (middle), and the matrix representation of the node features of the ILG (right).

In general, given a domain with predicate and function symbols $\Sigma_p$ and $\Sigma_n$, we have that there are $|\Sigma_V| = 5 + 3|\Sigma_p| + |\Sigma_n| + |\text{constant\_objects}|$ categorical node features representing the semantics of a node. Continuous node features indicate the value of numeric variables and the error of the expression in $s_0$ of unachieved numeric goals, and are set to zero for any other node.

## 3.2 The CCWL algorithm for numeric planning

The WL algorithm [WL68] has been adapted to computing features for graphs with categorical node attributes by [SSVL$^+$11]. A variant of the WL algorithm for graphs with continuous node attributes has been proposed by [TGL$^+$19] for the purpose of computing kernels with the Wasserstein distance between graph embeddings. However, the graph embeddings themselves are not invariant to the order of graphs in the nodes. Furthermore, from [CTT24b], non-linear kernels result in poorer generalisation compared to linear models in the context of L4P due to overfitting to the range of training targets. Morris et al. [MKKM16] constructed kernels for continuous node attributes by hashing Euclidean embeddings into categorical features but such a method loses the semantic meaning of numbers. Thus, we propose a new variant of the WL algorithm for graphs with both categorical and continuous node attributes for generating graph embeddings (CCWL algorithm). This algorithm is summarised in Alg. 1 and also depicted in Fig. 3.

---

**Algorithm 1: CCWL algorithm**

**Data:** A graph $\mathbf{G} = \langle \mathbf{V}, \mathbf{E}, \mathbf{F}_{\text{cat}}, \mathbf{F}_{\text{con}}, \mathbf{L} \rangle$ with continuous and categorical attributes, a deterministic and injective HASH function, allowed colours $\mathbf{C} = [\![|\mathbf{C}|]\!]$, a pooling function POOL, and number of CCWL iterations $L$.

**Result:** Feature vector of size $\mathbb{R}^{(1+d)|\mathbf{C}|}$.

1 $\kappa^0(v) \leftarrow \mathbf{F}_{\text{cat}}(v), \forall v \in \mathbf{V}$
2 **for** $j \in [\![L]\!]$ **do for** $v \in \mathbf{V}$ **do**
3 $\quad \lfloor \kappa^j(v) \leftarrow \text{HASH}\left(\kappa^{j-1}(v), \bigcup_{\iota \in \Sigma_E}\{(\kappa^{j-1}(u), \iota) \mid u \in \mathbf{N}_\iota(v)\}\right)$
4 $\mathbf{M} \leftarrow \bigcup_{j \in \{0\} \cup [\![L]\!]}\{\!\{\kappa^j(v) \mid v \in \mathbf{V}\}\!\}$
5 $\vec{v}_{\text{cat}} \leftarrow \left[\text{COUNT}(\mathbf{M}, c_1), \ldots, \text{COUNT}(\mathbf{M}, c_{|\mathbf{C}|})\right]$
6 $\mathbf{S}_i = \{v \mid v \in \mathbf{V} \text{ s.t. } \exists j \in \{0\} \cup [\![L]\!], \kappa^j(v) = c_i\}, \forall i \in \mathbf{C}$
7 $\vec{v}_{\text{con}} \leftarrow [\text{con}(1) \| \ldots \| \text{con}(|\mathbf{C}|)], \text{con}(i) = \text{POOL}_{v \in \mathbf{S}_i}(\mathbf{F}_{\text{con}}(v))$
8 **return** $\vec{v}_{\text{cat}} \| \vec{v}_{\text{con}}$

---

iter 0

$\alpha$

$\beta$

$\beta$ $\gamma$

1 [0.2, 0.4]

1 [0.2, 0.4]   1 [0.3, 1.3]   2 [0.5, 1.5]

iter 1

3

3   4   5

HASH$(1, \{(1, \alpha), (1, \beta)\}) = 3$
HASH$(1, \{(1, \beta), (2, \gamma)\}) = 4$
HASH$(2, \{(1, \gamma)\}) = 5$

con$(1) = [0.7\ 2.1]$, con$(2) = [0.5\ 1.5]$
con$(3) = [0.4\ 0.8]$, con$(4) = [0.3\ 1.3]$

$\underbrace{[3\ 1\ 2\ 1}_{\vec{v}_{\text{cat}}}\ \underbrace{0.7\ 2.1\ 0.5\ 1.5\ 0.4\ 0.8\ 0.3\ 1.3]}_{\vec{v}_{\text{con}}}$

Figure 3: CCWL with one iteration, POOL $= \sum$, and $\mathbf{C} = [4]$.

Lines 1–3 of Alg. 1 are the original steps of the WL algorithm for generating graph embeddings by iteratively refining categorical node features, which we call colours, with two differences. Firstly, we replaced the multi-set with a set in the input of the hashing function. This is because in planning, unseen colours arise from graphs with increasing degrees which occur for out-of-distribution testing problems of increasing size. This problem is limited by relaxing the hash input with a set, which trades expressivity for generalisation. Secondly, we make use of edge labels in the hashing function.

Lines 4–5 collect the counts of allowed colours $\mathbf{C}$ seen during the main loop of the algorithm to generate the categorical feature vector in the form of a histogram. We assume by relabelling colours that $\mathbf{C} = [\![|\mathbf{C}|]\!]$. Lines 6–7 generate features from pooling the continuous attributes from different

groups of nodes. More specifically, for each colour $c \in \mathbf{C}$, we find the set of nodes which have been assigned the colour $c$ some time during the refinement process and pool the continuous attributes of these nodes. Thus, we have $|\mathbf{C}|$ pooled continuous feature vectors which we concatenate together. We note that this pooling and concatenation process is invariant to the order of nodes in a graph in contrast to the intermediate graph embeddings generated for Wasserstein WL graph kernels by Togninalli et al. [TGL$^+$19]. The algorithm returns the concatenation of the categorical and continuous feature vectors as the final feature vector output for the graph in Line 8. We note that $d = 1$ when running CCWL on the $\nu$ILG representation of a numeric planning task.

We note that a drawback of the algorithm is that continuous attributes are not refined directly. This could be done by introducing one or more aggregation functions as parameters to the algorithm and refining continuous attributes by concatenating the aggregations of their neighbouring attributes. However, this method introduces an increase in the size of the continuous feature vector exponential in the number of layers, with base equal to the number of aggregation functions chosen. Moreover, we noted from informal experiments that this method led to overfitting of models to a large number of blended continuous features that do not have an obvious relation to the learning target.

Assuming a constant time hashing function, the complexity of the CCWL algorithm is $O(nL(\delta + d))$ where $n = |V|$ of the input graph, $\delta = \max_{u \in \mathbf{V}} \sum_{\iota \in \mathbf{L}} |\mathbf{N}_\iota(u)|$ is the degree of the graph, $d$ is the dimension of the continuous node attributes, and $L$ is the number of layers. The main computation comes from Line 3 which is performed $nL$ times and the hashing function takes an input of size $\delta$. Collecting the categorical feature vector takes the same time, while collecting the continuous feature vector takes $O(nLd)$ time. For reasonably sized $d \lesssim \delta$, as in the case of $\nu$ILG where $d = 1$, this is the same complexity as the original WL algorithm for generating graph features, which is $O(nL\delta)$.

## 4 Relational neural networks for numeric planning

Deep learning architectures such as graph neural networks (GNNs) [SGT$^+$09, GSR$^+$17] benefit in generating latent representations automatically with backpropagation when trained end-to-end [LBH15]. GNNs also benefit from being able to train and evaluate on arbitrary sized graphs. However, it is generally understood that the expressive power of GNNs is limited by the WL-algorithm and counting logics with two variables [XHLJ19, BKM$^+$20]. This result translates to the impossibility result of GNNs not being able to learn features that can work well for arbitrary planning domains [SBG22, CTT24a]. Nevertheless, their application to numeric planning tasks, in which both logical and numeric reasoning is required, is less well understood. Thus, we still propose GNNs as an additional baseline for L4NP and empirically evaluate their performance for numeric planning in Sec. 6.

For our GNN architecture, we perform a transformation on the node features of the $\nu$ILG from 3.1 as input for GNNs that can handle edge labels. More specifically, given a $\nu$ILG $\mathbf{G} = \langle \mathbf{V}, \mathbf{E}, \mathbf{F}_{\text{cat}} : \mathbf{V} \to \Sigma_\mathbf{V}, \mathbf{F}_{\text{con}} \to \mathbb{R}, \mathbf{L} \rangle$, we construct a new graph $\mathbf{G}'$ with continuous node attributes $\mathbf{X} : \mathbf{V} \to \mathbb{R}^{|\Sigma_\mathbf{V}|+2}$ defined by $\mathbf{X}(u) = \text{OH}(\mathbf{F}_{\text{cat}}(u)) \| [r_1, r_2]$, where $\text{OH}(\mathbf{F}_{\text{cat}}(u)) \in \{0, 1\}^{|\Sigma_\mathbf{V}|} \subseteq \mathbb{R}^{|\Sigma_\mathbf{V}|}$ denotes a one-hot encoding of the categorical node feature of $u$, and $r_1$ denotes the numerical value of numeric variable nodes defined by $r_1 = [u]^{s_0}$ if $u \in X_n(s_0)$ and $r_1 = 0$ otherwise, and $r_2$ denotes the goal error for numeric goal nodes defined by $r_2 = [u]^{s_0}$ if $u \in G_n$ and $r_2 = 0$ otherwise. We denote the $\nu$ILG for GNNs by $\langle \mathbf{V}, \mathbf{E}, \mathbf{X}, \mathbf{L} \rangle$ with notation for categorical features removed. Thus, we can use this graph encoding of numeric planning tasks as input into any downstream GNN that can handle edge labels or features.

Fig. 2 illustrates the node feature matrix representation of the $\nu$ILG encoding of a ccBlocksworld task for input to a GNN. Each row represents a node in the graph, with columns representing the semantics of the node as well as the value of the numeric variables in the state and error of numeric goal nodes. We note however, that the ccBlocksworld example does not have any numeric goals and thus the last column is zero for all entries.

## 5 Optimisation formulations for learning heuristic functions

In this section, we describe two optimisation methods used for learning heuristic functions from training data, namely by minimising cost-to-go estimate error and ranking estimate error. Fig. 4 illustrates examples of learned heuristic functions on states of a planning task when trained to zero loss with both the cost-to-go and ranking formulations. We assume that training data for our models consist of a set of numeric planning tasks $\Pi_1, \ldots, \Pi_n$ with corresponding optimal plans $\pi_1, \ldots, \pi_n$.

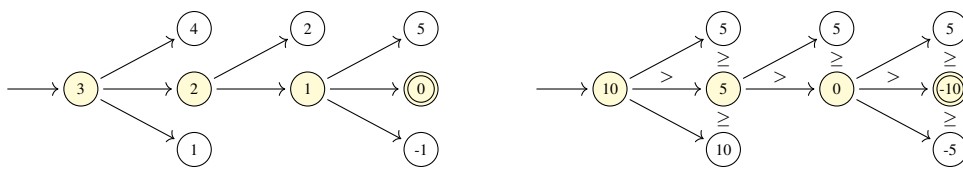

Figure 4: Examples of heuristic functions that achieve 0 loss when optimising cost-to-go (left) and ranking (right) on an optimal plan. Coloured nodes indicate states on the optimal plan with the goal state indicated by a double circle. Edges indicate successors of a node. A cost-to-go heuristic can achieve 0 loss on the plan trace but may not generalise correctly to state successors. A ranking heuristic does not need represent correct cost-to-go values and only need to satisfy ranking constraints. GBFS will return a plan in linear time for the ranking heuristic here but not for the cost-to-go heuristic.

We note that a numeric planning task can offer more training data by generating additional tasks and plans from different states in the state space of the task. Each plan is denoted $\pi_i = a_1^{(i)}, \ldots, a_{|\pi_i|}^{(i)}$ with plan trace $s_0^{(i)}, s_1^{(i)}, \ldots, s_{|\pi_i|}^{(i)}$. Each state $s$ in a plan trace induces a new planning task by replacing $s_0$ with $s$ of the original task, with which we can construct graph or vector representations from our aforementioned models.

**Heuristic functions from cost-to-go estimates.** We can use planning tasks and corresponding optimal plans as training data for learning a heuristic function representing the estimated cost-to-go to the plan. Each task and corresponding plan $\pi_i$ contributes training data $s_j^{(i)}$ with targets $h^*(s_j^{(i)})$ for each state $s_j^{(i)}$ in the plan trace of $\pi_i$. Then given an estimator $\mathcal{H}$, we may try to find weights that minimise the mean squared error (MSE) $L(\theta) = \frac{1}{N} \sum_{i=1}^{n} \sum_{j=0}^{|\pi_i|} \big( h^*(s_j^{(i)}) - \mathcal{H}_\theta(s_j^{(i)}) \big)^2$ where $N$ is the normalisation constant and $\mathcal{H}_\theta$ denotes the estimator with weights $\theta$.

**Heuristic functions from ranking estimates.** The MSE loss is a simple but naive method for training a heuristic function. Various researchers have instead proposed to use the concept of ranking to learn heuristic functions [GKL16, CEKP23, HTT$^+$24]. However, a drawback of the formulation of the ranking optimisation of previous works is that a state in a plan trace is marked as strictly better as its siblings when it could be the case that the siblings may have the same $h^*$ value. Furthermore, the formulation in [CEKP23] scales quadratically in the plan trace. We offer a novel ranking optimisation criterion that (1) fixes the problem of siblings being misclassified and (2) also results in a sparse model. We also offer a corresponding differentiable loss function for use with any end-to-end model.

Our first ranking formulation requires solving an LP as the optimisation problem, similarly to [FCGP19] but only using states from the plan trace, whereas the latter work uses states from the entire state space of the problem. It can also be viewed as an LP encoding of the formulation by Garrett et al. [GKL16] but fixing the problem of misrepresented siblings and learning sparse weights. Let $\text{SUCCS}(s)$ denote the set of successors of the state $s$ in a planning task by applying all applicable actions at $s$. Hence the set of siblings of state $s_j^{(i)}$ in $\Pi_i$'s state space is $\text{SIBLINGS}(s_j^{(i)}) = \text{SUCCS}(s_{j-1}^{(i)}) \setminus \{s_j^{(i)}\}$. Let $\varphi$ denote our feature generation function with $\varphi(s) \in \mathbb{R}^d$ for any state $s$. Then we can define our optimisation problem as a linear program defined by

$$\min_{\mathbf{w}, \mathbf{z}} \sum_{i,j,k} \mathbf{z}_{i,j,k} + \|\mathbf{w}\|_1 \quad \text{s.t.} \qquad \mathbf{z}_{i,j,k} \geq 0, \forall i, j, k \tag{1}$$

$$\mathbf{w}^\top (\varphi(s_{j-1}^{(i)}) - \varphi(s_j^{(i)})) \geq \text{cost}(a_j^{(i)}) - \mathbf{z}_{i,j,0} \qquad \forall i \in [\![n]\!], j \in [\![|\pi_i|]\!]$$

$$\mathbf{w}^\top (\varphi(s_\alpha) - \varphi(s_j^{(i)})) \geq -\mathbf{z}_{i,j,\alpha} \qquad \forall i \in [\![n]\!], j \in [\![|\pi_i|]\!], s_\alpha \in \text{SIBLINGS}(s_j^{(i)}).$$

The vector $\mathbf{w}$ represents the weights our linear model aims to learn, and the nonnegative slack variables $\mathbf{z}$ model the soft inequality constraints representing the ranking of states. The optimisation problem is to minimise the the slack variables corresponding to the error of the constraints, and the $\ell_1$ norm of the weights to encourage sparsity.

We next offer a differentiable loss function version of the previous model which we can use as a fair comparison when combining it with our GNN architecture in Sec. 4 compared to combining (1) with features generated in Sec. 3. The idea is to replace the slack variables with the $\max$ function:

$$L(\theta) = \sum_{i,j} \Big( \max\big(0, \mathcal{H}_\theta(s_j^{(i)}) - \mathcal{H}_\theta(s_{j-1}^{(i)}) + c(a_j^{(i)})\big) + \sum_{s_\alpha \in \text{SIBLINGS}(s_j^{(i)})} \max \Big(0, \mathcal{H}_\theta(s_j^{(i)}) - \mathcal{H}_\theta(s_\alpha)\Big)\Big). \tag{2}$$

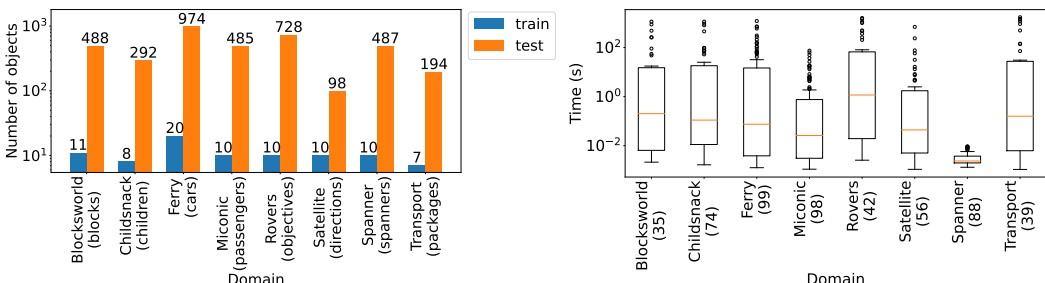

Figure 5: Number of objects in training and testing problems (left) and distributions of training data generation time with number of training problems (right) per domain. Note the log scales.

## 6 Experiments

### 6.1 Numeric planning benchmarks

We take 8 domains out of 10 domains from the International Planning Competition 2023 Learning Track (IPC-LT) [SSA23] and either convert them to equivalent numeric formulations, or introduce numeric variables to model extra features such as capacity constraints. The two domains from the IPC-LT that we do not convert into numeric domains are Floortile and Sokoban which do not have any benefit from compilation to a numeric representation nor exhibit any interesting features that can be modelled with numeric variables. The domains we considered from the IPC-LT are summarised in Fig. 5 alongside the sizes of training and testing tasks, and time to generate training data. Each domain consists of 90 testing problems and at most 99 small training problems for which the median time for generating an optimal training plan is less than a second and a few outliers taking more than a minute. We refer to the appendix for further details on the domains.

### 6.2 Experimental setup

**Training.** As discussed in Sec. 5, we only consider optimal plans from small problems as training data. We compute them with the Numeric Fast Downward planner [AN17] using $A^*$ search and the admissible $h^{\text{LMCUT}}$ heuristic [KSP+22], with a 30 minute timeout and 8GB main memory.

We consider 4 model configurations. Firstly, we use CCWL features from Sec. 3 with Support Vector Regression and the linear dot product kernel to learn a linear model for cost-to-go estimation ($h^{\text{CCWLF}}_{\text{cost}}$). Next, we use CCWL features in optimisation problem in (1) with CPLEX version 22.11 and a timeout of 600 seconds for ranking estimation ($h^{\text{CCWLF}}_{\text{rank}}$). Both $h^{\text{CCWLF}}_{\text{cost}}$ and $h^{\text{CCWLF}}_{\text{rank}}$ models have allowed colours $\mathbf{C}$ in Alg. 1 given by all the refined colours seen during training. We also have cost-to-go ($h^{\text{GNN}}_{\text{cost}}$) and ranking ($h^{\text{GNN}}_{\text{rank}}$) estimation models using GNNs operating on $\nu$ILG representations of planning tasks and optimised with the MSE loss function and (2), respectively. For the backbone GNN, we use a Relational Graph Convolution Network [SKB+18] but replacing the mean aggregation function with the element-wise max operator in the message-passing update step: $\mathbf{h}_u^{(l+1)} = \sigma(\mathbf{W}_0 \mathbf{h}_u^{(l)} + \sum_{\iota \in \Sigma_{\text{E}}} \max_{v \in \mathbf{N}_\iota(u)} \mathbf{W}_\iota^{(l)} \mathbf{h}_v^{(l)})$, where $l$ denotes the GNN layer, $\sigma$ is implemented with the leaky ReLU function, and $\mathbf{W}_0$ and $\mathbf{W}_\iota^{(l)}$ are learnable weight matrices. Each GNN has a hidden dimension of 64, and is trained with the Adam optimiser [KB15] with an initial learning rate of $10^{-3}$ and batch size of 16. A scheduler reduces the training loss by a factor of 10 if loss does not improve after 10 epochs. Training then terminates if the learning rate falls below $10^{-5}$. Let $L$ denote the iterations hyperparameter for CCWL models and number of layers for GNN models.

**Evaluation.** We consider several numeric planners as baselines for benchmarking the effectiveness of learning. We first include $h^{\text{LMCUT}}$ as the only optimal planner baseline as it is also the training data generator but solves a more difficult problem of optimal planning compared to satisficing planning. We consider the Metric-FF planner (M-FF) [Hof03], and the $h^{\text{ADD}}$, $h^{\text{MRP}}$, $h^{\text{MRP}}$+hj and $\text{M}(3h\|3n)$ configurations in the ENHSP planner [SHTR20, SSSG20, CT24]. We have that $h^{\text{ADD}}$ and $h^{\text{MRP}}$ are planners that perform GBFS with a single heuristic only, while $h^{\text{MRP}}$+hj and $\text{M}(3h\|3n)$ use additional techniques (macro actions, multiple queues, and novelty heuristics) to boost planning performance. Our CCWL and GNN models are all used in single-queue GBFS with the learned heuristic function, with Numeric Fast Downward as the backend search implementation. All baselines and models are run on a single Intel Xeon Platinum 8268 (2.90 GHz) core with a 5 minute timeout for search and

Table 1: Coverage of numeric domain-independent, the new learning planners ($h_{\text{cost}}^{\text{GNN}}$, $h_{\text{rank}}^{\text{GNN}}$, $h_{\text{cost}}^{\text{CCWLF}}$, $h_{\text{rank}}^{\text{CCWLF}}$) with $L = 1$, and the best learner configuration score on each domain (Best Learner). Higher values are better (↑), with the top three scores in each row except the rightmost entry indicated by the cell colour intensity. All planner configurations except $h_{\text{opt}}^{\text{LMCUT}}$ are satisficing planners.

| | Planner Baselines | | | | | | Learners (new) | | | | |
| | | | | | | | GBFS + heuristic | | | | |
| Domain | $h_{\text{opt}}^{\text{LMCUT}}$ | $h^{\text{MRP}}+\text{hj}$ | M-FF | $\text{M}(3h\|\|3n)$ | $h^{\text{MRP}}$ | $h^{\text{ADD}}$ | $h_{\text{cost}}^{\text{GNN}}$ | $h_{\text{rank}}^{\text{GNN}}$ | $h_{\text{cost}}^{\text{CCWLF}}$ | $h_{\text{rank}}^{\text{CCWLF}}$ | Best Learner |
|---|---|---|---|---|---|---|---|---|---|---|---|
| Blocksworld | 6 | 18 | 9 | 23 | 19 | 16 | 18 | **24** | 22 | 19 | **29** |
| Childsnack | 20 | 49 | 14 | 53 | 25 | 20 | 17 | 22 | 22 | **90** | 90 |
| Ferry | 33 | 60 | 60 | 57 | 60 | 57 | 60 | 60 | 70 | **71** | 73 |
| Miconic | 30 | 68 | 65 | 61 | 64 | 51 | 63 | 64 | **90** | **90** | 90 |
| Rovers | 10 | **34** | 17 | 30 | 18 | 15 | 18 | 14 | 22 | 23 | 30 |
| Satellite | 18 | **38** | 24 | 29 | 21 | 23 | 19 | 14 | 23 | 16 | 26 |
| Spanner | 30 | 6 | 35 | 76 | 42 | 42 | 90 | 90 | 90 | 90 | 90 |
| Transport | 12 | **55** | 49 | 40 | 32 | 40 | 34 | 38 | 40 | 46 | 48 |
| Σ | 159 | 328 | 273 | 369 | 281 | 264 | 319 | 326 | 379 | **445** | 476 |

8GB of main memory. Tab. 1 summarises the coverage results of all considered planners on the benchmarks, with more details provided in the appendix.

**How do learning approaches compare to domain-independent numeric planners?** From Tab. 1, we note that our best performing model with $L = 1$ is $h_{\text{rank}}^{\text{CCWLF}}$ and outperforms all domain-independent planners for satisficing planning on 4 out of 8 domains. Increasing $L$ to 2 brings $h_{\text{rank}}^{\text{CCWLF}}$ to achieve the best coverage on Blocksworld. The domains which learners fall behind on are Rovers, Satellite and Transport, even when taking the best hyperparameter configuration. The former two are difficult as they require features more expressive than those generated by graph learning approaches to capture the semantics of reasoning required to solve the problems [SBG22], while the latter requires path finding which is not possible for learners with finite receptive fields [TTTX20]. These results hold for classical planning and thus also for our extension to numeric planning. Generally the best performing planner on a domain expands fewer nodes than the other planners. With regards to plan length, $h_{\text{rank}}^{\text{CCWLF}}$ performs best for Blocksworld but is marginally worse than the best of the domain-independent numeric planners for Rovers, Satellite and Spanner.

**How do CCWL models compare to GNN models?** From Tab. 1, we see that the CCWL models always have similar or better performance than the corresponding GNN models, when comparing cost-to-go and ranking estimates. The performance of a planners which use GBFS and a heuristic depend on the heuristic evaluation speed, in which more search can be done in the time limit, or the quality of the heuristic, in which search can be more informed. Fig. 8 in the appendix shows that GNN are generally at least an order of magnitude slower than CCWL models for heuristic evaluation due to performing intensive matrix operations. We note that GNN models are evaluated on CPUs and could be sped up with access GPUs. Fig. 6a illustrates the number of node expansions of GNN and CCWL models and we note that there is no clear winner between the two approaches across all domains, with the exception of $h_{\text{rank}}^{\text{CCWLF}}$ generalising perfectly on Childsnack where other models could not. Thus, we can conclude with respect to planning efficiency that CCWL models generally outperform their GNN counterparts due to faster heuristic evaluation speeds, while generally both models have similar generalisation performance.

**How do ranking models compare to cost-to-go models?** From Tab. 1, ranking models outperform cost-to-go models in total coverage. However, their performance is incomparable across domains even when looking at Fig. 6b with the exception of CCWL being able to achieve perfect performance on Childsnack. Nevertheless, on 8 domain-model pairs for $L = 1$, ranking models achieve strictly better coverage, while the converse is only true for 4 domain-model pairs. This suggests a bias favouring ranking models which can be explained by their advantages covered in Sec. 5, namely that

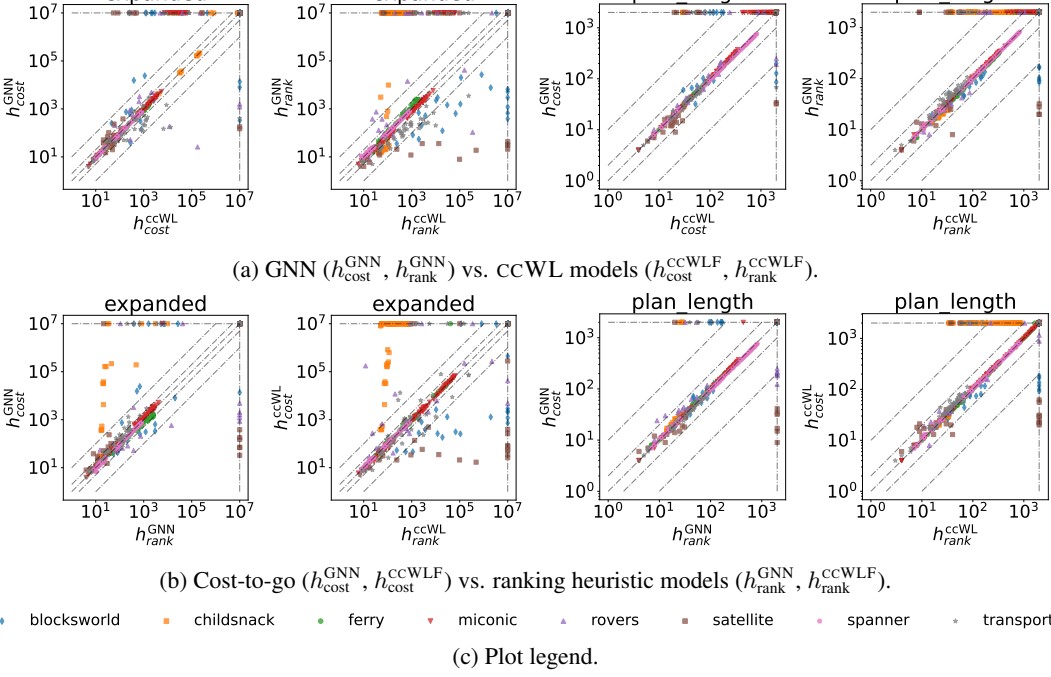

(a) GNN ($h_{\mathrm{cost}}^{\mathrm{GNN}}$, $h_{\mathrm{rank}}^{\mathrm{GNN}}$) vs. CCWL models ($h_{\mathrm{cost}}^{\mathrm{CCWLF}}$, $h_{\mathrm{rank}}^{\mathrm{CCWLF}}$).

(b) Cost-to-go ($h_{\mathrm{cost}}^{\mathrm{GNN}}$, $h_{\mathrm{cost}}^{\mathrm{CCWLF}}$) vs. ranking heuristic models ($h_{\mathrm{rank}}^{\mathrm{GNN}}$, $h_{\mathrm{rank}}^{\mathrm{CCWLF}}$).

♦ blocksworld   ■ childsnack   ● ferry   ▼ miconic   ▲ rovers   ■ satellite   ● spanner   ✳ transport

(c) Plot legend.

Figure 6: Plot comparisons of expanded nodes and plan length of selected pairs of models with $L = 1$. A point $(x, y)$ represents the metric of the models indicated on the $x$ and $y$ axis on the domain. Points on the top left (resp. bottom right) triangle favour the model on the $x$-axis (resp. $y$-axis).

they implicitly use more training data by considering successor states of plan trace states, and have a larger solution space as they are not restricted to predicting an exact value.

**What is the effect of number of iterations for CCWL models and layers for GNNs?** The hyperparameter $L$, which denotes the number of iterations (resp. layers) for CCWL (resp. GNN) models, generally plays an important role in planning performance. This is because increasing $L$ improves model expressivity and reasoning capabilities, but comes at the cost of heuristic evaluation time and increased possibility of overfitting to the training data. From Tab. 2 in the appendix, we note that surprisingly for most domains and models $L = 0$ or $L = 1$ provides the best coverage, while increasing $L$ rarely improves coverage. This suggests that heuristic evaluation time plays an important role in planning performance for domains that cannot be solved with the learner's expressivity.

## 7   Limitations

The setup of our work is limited to the assumption that the problems being solved can be explicitly represented in a symbolic language such as PDDL. The assumption of the existence of PDDL encodings of planning problems allows us to generate training data quickly with domain-independent numeric planners for supervised training. Furthermore, experiments and theoretical insights also show that our proposed techniques have room for improvement as there are still classes of numeric planning tasks with which our models cannot learn and generalise well in.

## 8   Conclusion

We have proposed a new graph embedding algorithm, the CCWL algorithm, and optimisation criterion for learning heuristic functions for numeric planning. Planning tasks are encoded as Numeric Instance Learning Graphs ($\nu$ILG) on which we run our CCWL algorithm for generating features. Our numeric planning features are interpretable and efficient to generate. Experimental results show the effectiveness of our approach by achieving competitive performance over both deep learning architectures and domain-independent numeric planners. Furthermore, we have identified future work by improving the expressivity of our algorithms for capturing more complex numeric domains. Lastly, one can learn forms of domain knowledge different from heuristic functions with our new numeric planning features and graph representations such as policies [WT24], portfolios [MFH+20] and detecting relevant objects [SCC+21].

## Acknowledgements

The authors thank the reviewers and Giacomo Rosa for the helpful comments and suggestions. The computing resources for the project was partially supported by the Australian Government through the National Computational Infrastructure (NCI) under the ANU Startup Scheme. ST was supported by the Australian Research Council grant DP220103815, by the Artificial and Natural Intelligence Toulouse Institute (ANITI) under the grant agreement ANR-23-IACL-0002, and by the European Union's Horizon Europe Research and Innovation program under the grant agreement TUPLES No. 101070149.

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

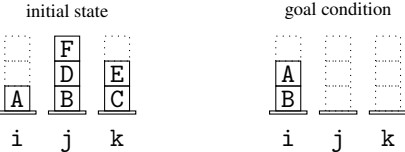

Figure 7: Left: initial state of a ccBlocksworld problem, where base `i`, `j`, and `k` each have a load limit of 3 blocks. Right: the goal condition where `A` is on top of `B` which is on top of `i`.

## A  More Details for the ccBlocksworld Example

We repeat the running ccBlocksworld example in Fig. 7. Listings 2 and 1 provide the explicit PDDL domain and problem encodings for the running ccBlocksworld example. An optimal plan for the problem is given as follows on the left, and an optimal plan without capacity constraints on the right.

| | | | |
|---|---|---|---|
| 1. | (unstack f d j) | 1. | (unstack f d j) |
| 2. | (stack f a i) | 2. | (stack f e k) |
| 3. | (unstack d b j) | 3. | (unstack d b j) |
| 4. | (stack d f i) | 4. | (stack d f k) |
| 5. | (pickup b j) | 5. | (pickup a i) |
| 6. | (stack b e k) | 6. | (stack a d k) |
| 7. | (unstack d f i) | 7. | (pickup b j) |
| 8. | (putdown d j) | 8. | (putdown b i) |
| 9. | (unstack f a i) | 9. | (unstack a d k) |
| 10. | (stack f d j) | 10. | (stack a b i) |
| 11. | (pickup a i) | | |
| 12. | (stack a f j) | | |
| 13. | (unstack b e k) | | |
| 14. | (putdown b i) | | |
| 15. | (unstack a f j) | | |
| 16. | (stack a b i) | | |

Listing 1: PDDL encoding for the ccBlocksworld problem in Fig. 7.

```
(define (problem running-example)
    (:domain ccblocksworld)
    (:objects
      a b c d e f - block
      i j k - base
    )
    (:init
        (arm_empty)
        (= (capacity i) 2)
        (= (capacity j) 0)
        (= (capacity k) 1)
        (clear a)
        (on a i)
        (above a i)
        (clear f)
        (on f d)
        (on d b)
        (on b j)
        (above f j)
        (above d j)
        (above b j)
        (clear e)
        (on e c)
        (on c k)
        (above e k)
        (above c k)
    )
    (:goal (and
        (clear a)
        (on a b)
        (on b i)
    ))
)
```

Listing 2: PDDL encoding for the ccBlocksworld domain.

```
(define (domain ccblocksworld)
    (:requirements :strips :typing :numeric-fluents)
    (:types
        block base - object
    )
    (:predicates
        (on          ?x - block        ?y - object)
        (above       ?x - block        ?y - base  )
        (clear       ?x - object                  )
        (holding     ?x - block                   )
        (arm_empty                                )
    )
    (:functions
        (capacity    ?x - base                    )
    )
    (:action pickup
        :parameters (?block - block ?base - base)
        :precondition (and
            (on ?block ?base)
            (above ?block ?base)
            (clear ?block)
            (arm_empty))
        :effect (and
            (not (on ?block ?base))
            (not (above ?block ?base))
            (not (clear ?block))
            (clear ?base)
            (holding ?block)
            (not (arm_empty))
            (increase (capacity ?base) 1))
    )
    (:action putdown
        :parameters (?block - block ?base - base)
        :precondition (and
            (holding ?block)
            (clear ?base)
            (<= 1 (capacity ?base)))
        :effect (and
            (not (holding ?block))
            (not (clear ?base))
            (on ?block ?base)
            (above ?block ?base)
            (clear ?block)
            (arm_empty)
            (decrease (capacity ?base) 1))
    )
    (:action unstack
        :parameters (?block_a - block ?block_b - block ?base - base)
        :precondition (and
            (on ?block_a ?block_b)
            (above ?block_a ?base)
            (clear ?block_a)
            (arm_empty))
        :effect (and
            (not (on ?block_a ?block_b))
            (not (above ?block_a ?base))
            (not (clear ?block_a))
            (clear ?block_b)
            (holding ?block_a)
            (not (arm_empty))
            (increase (capacity ?base) 1))
    )
    (:action stack
        :parameters (?block_a - block ?block_b - block ?base - base)
        :precondition (and
            (holding ?block_a)
            (clear ?block_b)
            (above ?block_b ?base)
            (<= 1 (capacity ?base)))
        :effect (and
            (not (holding ?block_a))
            (not (clear ?block_b))
            (on ?block_a ?block_b)
            (above ?block_a ?base)
            (clear ?block_a)
            (arm_empty)
            (decrease (capacity ?base) 1))
    )
)
```

# B    Related Work

Two related fields to Learning For Planning (L4P) and Learning For Numeric Planning (L4NP) are Generalised Planning (GP) and Reinforcement Learning (RL). In the following subsections, we outline the main difference between L4P with the respective related fields as well as corresponding related work.

## B.1    Generalised planning

GP consists of automatically characterising the solution of a (possibly infinite) set of planning tasks [Sri10, SIZ08]. The most common characterisations are action policies, but other character-isations also include finite state controllers [BPG09, BPG10, HG11, HG13, AJJ18], and programs with branching and looping [AJJ21, ACSJ22]. Logic programming approaches involving decision lists [Kha99, GT04] and Datalog programs [GRH24, CHŠ24] have also been used to characterise solutions for planning domains. We refer to articles [CAJ19] and [Sri23] for more detailed surveys of GP. The difference L4P and GP can be subtle given that there is a non-empty intersection between the two fields, and works in both fields generally aim to compute structures that solve problems from a given domain. The way we differentiate the two fields is that L4P follows generally follows traditional supervised learning approaches, whereas GP can be likened to performing program synthesis.

With regards to numeric planning, Srivastava et al. [SZIG11] introduced Qualitative Numeric Planning (QNP) which is a subset of numeric planning where numeric variables exhibit non-negative domains, and actions increase or decrease the value of numeric variables by indeterminate amounts. A solution for a QNP is a policy which can be used to represent solutions for sets of planning tasks. QNP has been shown to be equivalent to fully observable non-deterministic (FOND) planning [BG20] arising from the non-determinism of action effects, and the connection between FOND and GP has often shown itself when used to synthesise generalised policies [BG18, IM19]. Lin et al. [LCF$^+$22] studies GP for a more expressive class of numeric planning, by allowing for integer numeric variables and employing linear expressions in conditions and action effects. Their approach involves synthesising programs that allow for branching and looping. Lastly, $\nu$ASNets [WT24] extends ASNets [TTTX20] in order to learn policies with a neural network architecture for planning.

## B.2    Reinforcement Learning

RL is a learning paradigm for decision making that does not have access to a model and instead learns from rewards [SB98]. RL has achieved promising results in games when combined with deep learning [MKS$^+$15, SHM$^+$16]. A major difference between RL and L4P is that the former requires reasoning over dense reward functions, whereas the latter requires reasoning over logic [Gef18]. Nevertheless, there has been some preliminary work looking at the intersection of RL and planning. Reward machines [IKVM22] are a logical language used for specifying reward functions for RL problems, inspired by the declarative nature of the planning as modelling paradigm. RL has also been applied directly into planning tasks, as done by [MV21] for temporal planning. Rewards are mostly sparse, with 1 being reward for achieved goals, minor $10^{-5}$ rewards for achieved goal propositions, and no reward otherwise. Gehring et al. [GAC$^+$22] explored introducing denser reward functions to planning through domain-independent heuristics to allow for RL approaches. Supervised RL has also been used for learning planning policies [SBG23]. Nevertheless, the use cases for RL and planning are generally different, with RL being more suited for control tasks in continuous or dynamic environments such as in robotics, and planning being more suited for combinatorial tasks in discrete or abstract environments such as in logistics.

# C    Description of Benchmark Domains

## C.1    Numeric (Capcity Constrained) Blocksworld

This domain was described in Sec. 2. A task from the domain consists of $n$ blocks stacked on top of one another to form towers on top of $b$ bases. Each base has a capacity of how many blocks it can support. The goal is to stack and unstack blocks to achieve a target tower configuration. The numeric component of this domain arises from modelling the capacity of bases. Training problems have $n \in [2, 11]$ blocks while testing problems have $n \in [5, 488]$ blocks.

## C.2    Numeric Childsnack

A task from the domain consists of feeding $c$ children with sandwiches in $l$ locations, of which some are allergic to gluten. There are a finite amount of gluten-free (GF) and non-GF ingredients.

GF sandwiches can only be made from GF ingredients, whereas non-GF sandwiches can be made with any ingredients. Children allergic to gluten are only allowed to eat GF sandwiches while the remaining children can eat any type of sandwich. Thus, the problem has deadends because resources are finite and can be wasted. The goal is to make sandwiches and feed all the children satisfying the aforementioned rules. The numeric component of the domain arises from modelling the ingredient and sandwich resources. Training problems have $c \in [1, 8]$ children while testing problems have $c \in [4, 292]$ children.

## C.3   Numeric Ferry

A task from the domain consists of $c$ cars spread across $l$ locations. A ferry is able to transport up to a fixed amount of cars around to different locations. The goal of the domain is to transport the cars with the ferry to various target locations. The numeric component of the domain arises from modelling the capacity of the ferry. Training problems have $c \in [1, 20]$ cars while testing problems have $c \in [4, 974]$ cars.

## C.4   Numeric Miconic

A task from the domain consists of $p$ passengers with different weights spread across $f$ floors. There is a single elevator with a fixed load capacity that can transport passengers between floors. Furthermore, if the load of the elevator exceeds a secondary threshold, it takes twice as long to move between floors. The goal of the domain is to move the passengers to their target floors. The numeric component of the domain arises from modelling the weight of the passengers and load capacity of the elevator. Training problems have $p \in [1, 10]$ passengers while testing problems have $p \in [1, 485]$ passengers.

## C.5   Numeric Rovers

A task from the domain consists of $r$ rovers some of which can sample rock and soil data, while others have cameras that can take images of objectives. The goal of each problem is to sample rock and soil data as well as take images of objectives and communicate all $g$ data to the lander. The rovers can move around a map with $w$ waypoints and the rover is only able to communicate data to the lander from a subset of waypoints. Furthermore, rovers have a limited energy supply that is consumed with any action, but they can recharge with solar panels at certain waypoints. Thus, the problem has deadends because rovers have limited energy and could exhaust them in waypoints where they cannot recharge. The numeric component of the domain arises from modelling the energy supply of the rovers. Training problems have $g \in [1, 10]$ goals while testing problems have $[2, 728]$ goals problems

## C.6   Numeric Satellite

A task from the domain consists of $s$ satellites, each carrying a subset of $i$ instruments that can take pictures of space using a subset of $m$ modes. Satellites can rotate to take pictures of $d$ locations in space. Each satellite has a fixed amount of fuel that is consumed when rotating, and a fixed amount of data capacity that is consumed when taking pictures. Thus, the problem has deadends because resources are finite and can be wasted. The goal of a Satellite problem is to take pictures of a set of locations in space with specified modes while adhering to the fuel and data capacity constraints. The numeric component of the domain arises from modelling the fuel and data capacity features. Training problems have $s \in 2, 10$ satellites and testing problems have $s \in [4, 98]$ satellites.

## C.7   Numeric Spanner

A task from the domain consists of $s$ spanners scattered along a one-way hallway with $l$ locations, and $n$ nuts at the end of the hallway that have to be fixed. Each spanner can only be used to fix a single nut before it breaks. The goal of the domain is to fix all the nuts. The problem has deadends if not enough spanners are picked up before reaching the end of the hallway. The numeric component of the domain arises from modelling the number of spanners and nuts. Training problems have $s \in [1, 10]$ spanners while testing problems have $s \in [1487]$ spanners.

## C.8   Numeric Transport

A task from the domain consists of $p$ packages spread across $l$ locations, with $t$ number of trucks that can transport pick up and transport packages on a map. Each truck has a limited capacity of packages it can carry. The goal of the problem is to transport all the packages to their target locations. The numeric component of the domain arises from modelling the capacity of the trucks. Training problems have $p \in [1, 7]$ packages while testing problems have $p \in [1, 194]$ packages.

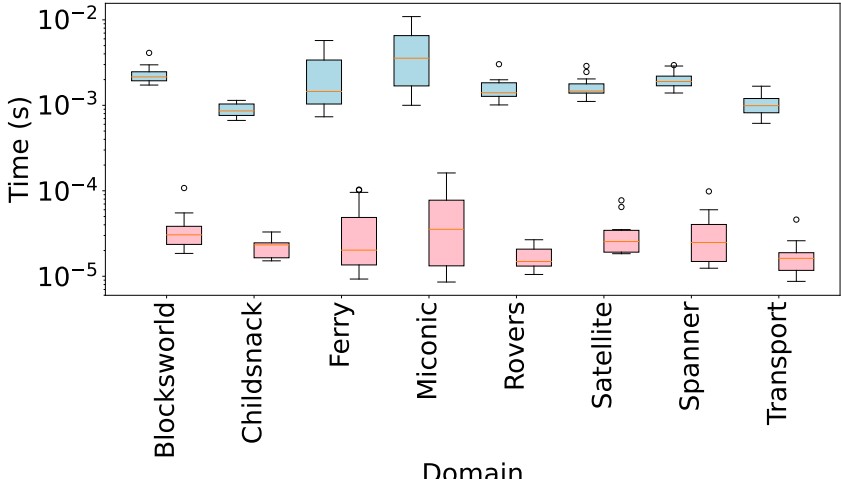

Figure 8: Distributions of heuristic evaluation time for GNN and CCWL models with $L = 1$ on problems where both were able to solve in the given timeout. Blue box plots correspond to GNN models and red box plots correspond to CCWL models.

## D    GNN and CCWL heuristic evaluation times

We refer to Fig. 8 for distributions of heuristic evaluation time for GNN and CCWL models with $L = 1$. Times are computed by taking the total search time for each problem and dividing by the number of heuristic evaluations made by the planner. We assume that the heuristic evaluation is the bottleneck of the search which is confirmed with informal profiling experiments.

## E    Effect of number of iterations and layers

We refer to Tab. 2 for the coverage of models with different $L$ values.

Table 2: Coverage and median of expansions of solved problems for each CCWL and GNN model with varying number of iterations and layers. Higher values are better for coverage, and lower values are better for expansions. The best value per domain and metric coloured. OOM denotes that the training process exceeded the memory limit.

(a) $h_{\text{cost}}^{\text{GNN}}$

| Domain | Coverage | | | | |
| | 0 | 1 | 2 | 3 | 4 |
|---|---|---|---|---|---|
| Blocksworld | 22 | 18 | **24** | 19 | 21 |
| Childsnack | **18** | 17 | 15 | 14 | 13 |
| Ferry | **60** | **60** | **60** | **60** | **60** |
| Miconic | **67** | 63 | 62 | 60 | 58 |
| Rovers | **21** | 18 | 11 | 13 | 13 |
| Satellite | **22** | 19 | 11 | 13 | 14 |
| Spanner | **90** | **90** | **90** | **90** | 77 |
| Transport | **36** | 34 | 31 | 31 | 30 |
| Σ | **336** | 319 | 304 | 300 | 286 |

(b) $h_{\text{rank}}^{\text{GNN}}$

| Domain | Coverage | | | | |
| | 0 | 1 | 2 | 3 | 4 |
|---|---|---|---|---|---|
| Blocksworld | 20 | **24** | 22 | **24** | 22 |
| Childsnack | 18 | 22 | 26 | 29 | **37** |
| Ferry | **60** | **60** | **60** | **60** | **60** |
| Miconic | **70** | 64 | 62 | 62 | 59 |
| Rovers | **22** | 14 | 13 | 12 | 11 |
| Satellite | **22** | 14 | 14 | 17 | 14 |
| Spanner | **90** | **90** | **90** | **90** | **90** |
| Transport | 36 | **38** | 33 | 27 | 28 |
| Σ | **338** | 326 | 320 | 321 | 321 |

(c) $h_{\text{cost}}^{\text{CCWLF}}$

| Domain | Coverage | | | | |
| | 0 | 1 | 2 | 3 | 4 |
|---|---|---|---|---|---|
| Blocksworld | **28** | 22 | 19 | 21 | 22 |
| Childsnack | **22** | **22** | 20 | 20 | 20 |
| Ferry | **73** | 70 | 65 | 62 | 58 |
| Miconic | **90** | **90** | 88 | 87 | 85 |
| Rovers | **30** | 22 | 19 | 15 | 15 |
| Satellite | **26** | 23 | 5 | 7 | 5 |
| Spanner | **90** | **90** | 89 | 88 | 86 |
| Transport | **48** | 40 | 37 | 30 | 27 |
| Σ | **407** | 379 | 342 | 330 | 318 |

(d) $h_{\text{rank}}^{\text{CCWLF}}$

| Domain | Coverage | | | | |
| | 0 | 1 | 2 | 3 | 4 |
|---|---|---|---|---|---|
| Blocksworld | 25 | 19 | **29** | 23 | 22 |
| Childsnack | 22 | **90** | 20 | 23 | 23 |
| Ferry | **73** | 71 | 70 | 68 | 68 |
| Miconic | **90** | **90** | 89 | 87 | 85 |
| Rovers | 21 | **23** | 15 | 22 | 21 |
| Satellite | **17** | 16 | 7 | OOM | OOM |
| Spanner | **90** | **90** | 89 | 89 | 89 |
| Transport | **48** | 46 | 46 | 29 | 35 |
| Σ | 386 | **445** | 365 | 341 | 343 |

