# OpenReview forum: "Graph Learning for Numeric Planning"
_NeurIPS.cc/2024/Conference — NeurIPS 2024 poster_

### Official Review · Reviewer_bZ6v · 2024-07-11

**Soundness:** 2
**Presentation:** 1
**Contribution:** 2
**Rating:** 3
**Confidence:** 3

**Summary:**

This paper proposed new learning-based methods for numeric planning. Numeric planning is formalized with the PDDL language. The proposed approaches are based on graph neural networks, and are evaluated in a lot of domains, e.g., blockworld, childsnack.

**Strengths:**

The experiment section seems solid, and the proposed approaches are evaluated in lots of domains. The experiment results demonstrate significant improvement over the compared baseline.

**Weaknesses:**

As there is only one closely related work on learning for numeric planning, it is hard to assess whether this is an important research problem.

The abstract has not summarized the novelty of this work, and the relationship between the two proposed approaches.

There are some missing related works about learning heuristics: evolution of Heuristics github.com/FeiLiu36/EoH.

**Questions:**

Are the proposed two approaches suitable for different scenarios? Can the authors analyze this issue?

**Limitations:**

This paper has discussed the potential limitations.

---

> ### Author Rebuttal · Authors · 2024-08-03
>
> We thank the reviewer for the comments and suggestion for related work.
>
> ## Questions. (Applicability to Different Scenarios)
> The proposed approaches are suitable for various different scenarios such as reinforcement learning (RL) and more general graph representation learning (GRL) problems.
>
> - RL
> Although we focused on the supervised learning setting in the paper, the proposed methods, namely the graphical representation of relational planning states and feature extraction algorithms, can be applied in the RL setting where a factored representation of states are given. More specifically, the graphical representation of the task and the GNN model/or simply a neural network operating on the features generated from the ccWL algorithm, can be applied with typical RL methodologies such as PPO and DQN. Furthermore, a model for the actions is not even needed and a simulator can even suffice as our methods only assume a symbolic representation of states and nothing about actions.
>
> - GRL
> We have also proposed a new graph kernel (ccWL kernel in Sec. 3.2) which can handle graphs with both continuous and categorical attributes. As mentioned in the paper, there are few graph kernels which handle non-categorical attributes and the ones that do are more statistical in nature. Our ccWL kernel is a new graph kernel that may be well suited for GRL tasks that require reasoning over logic such as knowledge graphs, as opposed to statistical reasoning as in common GRL benchmarks such as molecular datasets.
>
> ## Weaknesses
> W1: You question the importance of research on learning for numeric planning on the basis of the fact that there is only one other published work on this problem (published in 2024). However, the reason for there being few approaches so far, is not that the research problem is unimportant, but that it is *new* and *difficult*. There are a lot of approaches for numeric planning (including the baselines we use), and planning competitions for numeric planning being run regularly since 2002. However these works do not use machine learning at all.
>
> According to Turing Award winner Yann Le Cun, “learning to plan complex action sequences” is one of the top 3 challenges for deep learning  – see for instance his invited talks at AAAI 2020 (47:20 on the youtube video) and AAAI 2024 on “Objective-Driven AI: Towards AI systems that can learn, remember, reason, and plan”. The first works using modern deep learning techniques that obtained some success with classical (non-numeric) planning date back from 2018, and so are relatively recent. Numeric planning is the natural next step in this important line of work. As we show, learning has the potential to improve the state of the art in numeric planning.
>
> W2: We agree that the abstract can be improved and will do the suggested changes. Regarding the novelty of this work, this is the first work that learns heuristics for numeric planning – and one of the works in learning for planning that shows a clear improvement over the state-of-the-art. The current work on learning heuristics for planning is centered on classical planning where variables are Boolean.
>
> The two proposed approaches (Graph Neural Networks and Graph Kernels) are the two common classes of ML methods that operate on graph data. They have their own respective advantages and disadvantages akin to those of deep learning vs classical machine learning. More specifically, GNNs are more well suited at handling large data and extracting latent features via backprop for structured data, while graph kernels benefit more from small training sizes and are much more lightweight in terms of parameters. We proposed using both approaches to be comprehensive in our evaluation on our inputs transformed into graphs. We will make this relationship and motivation for using both approaches clear in our abstract.
>
>
> W3: The main common point between the work of Fei Liu and ours is the use of the word “heuristic” – yet with a different meaning.  In planning and heuristic search, the meaning of the word “heuristic” is very precise:  a heuristic is a function taking as input a state, and returning an estimate of the cost to reach the goal from this state.  The meaning of the word “heuristic” is different in other areas, such as optimisation, where it means “approximate decision”, “rule of thumb”, or “strategy”. The paper you mention is about meta-heuristic optimisation, which is a branch of optimisation where one seeks to design general-purpose strategies to guide the search towards an approximate solution. This is only very loosely related to our work. There is an abundance of recent work on learning branching heuristics for mixed-integer programming, and other types of optimisation solvers, which again target very different types of problems. We are happy to mention them in the final version, taking advantage of the extra page available.

---

### Official Review · Reviewer_99rS · 2024-07-12

**Soundness:** 3
**Presentation:** 1
**Contribution:** 3
**Rating:** 5
**Confidence:** 4

**Summary:**

The paper proposes a new method for learning a heuristic function to guide search for solving numeric planning problems. In contrast to classical planning, the states in numeric planning may involve numeric variables while the state transitions are defined by mathematical expressions over these kinds of variables. In addition, numeric planning is computationally quite challenging to solve but it may be the right formulation in many interesting real-world applications. The proposed heuristic is learned from training data consisting of example planning instances and their corresponding optimal plans. Features are automatically extracted from a graphical representation of the planning instances and used subsequently by the machine learning models encoding the heuristic functions. The experimental evaluation is carried out on standard benchmarks for numeric planning. The results show clearly that state-of-the-art search algorithms guided by the proposed heuristics improve considerably over their competitors.

**Strengths:**

- The paper targets an important problem in the area of automated planning and proposes more effective heuristics to guide search algorithms for solving these problems.

- The empirical evaluation is sound and covers standard benchmarks in the numeric planning domain. The results are presented in a relatively clear manner and therefore it is easy to understand the benefits of the proposed approach compared to existing state-of-the-art.

**Weaknesses:**

- In my opinion, the presentation is the main weakness of the paper. While I believe it is a solid contribution for numeric planning, the way it is currently presented makes it hard to understand the details.

- The examples currently supporting sections 3.1 and 3.2 need to be expanded a little more. Right now, it is kind of difficult to follow them.

- Section 4 which describes the graph neural network is too high level and without a supporting example is very hard to understand.

- There is currently a relatively big disconnect between Sections 3, 4 and 5. Namely it takes a while to figure out that the features extracted by the proposed method are subsequently used to learn the heuristics. The current notation doesn't help much either. I also think that adding illustrative examples in Section 5 would clearly improve the quality of the presentation.

**Questions:**

- The paper claims that grounding is not required for building the graphical representation of the planning problem. However, the graph shown in Figure seems to be partially grounded. So, how much grounding is needed to build such graphs?

**Limitations:**

The limitations are addressed clearly in the paper.

---

> ### Author Rebuttal · Authors · 2024-08-03
>
> We thank the reviewer for the comments and suggestions identifying which parts of the paper can be made clearer.
>
> ## Weaknesses
>
> We agree that the paper presentation can be improved and found it challenging to fit all of the material in the page limit. We will make good use of an extra page to address the presentation issues mentioned with more detail and illustrations. Please see the figure in the pdf of our global rebuttal for helping clarify some of our proposed methods as pointed in the final two points of your review.
>
>
> ## Question about Grounding
>
> The short answer to your question is that we do *no* grounding when encoding what is given to us, i.e. a lifted planning task (lines 84-85), to a graph. More specifically, the grounded items you see (green, yellow, and red nodes) in Figure 1 are the minimal amount of information required to define a planning task. This information includes the current state, and the goal condition.
>
> The choice of nomenclature of a “lifted planning task” is because our graphs need the first-order representation of the task, as opposed to the flattened representation, in order to be able to identify the object (blue nodes) and link them via edges to the predicates and functions that use them as arguments in the initial state and the goal.

---

> > ### Comment · Reviewer_99rS · 2024-08-10
> >
> > Thanks for the clarifications.

---

### Official Review · Reviewer_R2Rg · 2024-07-15

**Soundness:** 3
**Presentation:** 3
**Contribution:** 3
**Rating:** 7
**Confidence:** 4

**Summary:**

The paper tackles numeric planning problems by proposing two heuristics for numeric planning. The first one is based on graph kernels for graphs and addresses both continuous and categorical attributes. The second uses graph neural networks. The authors experimentally show the effectiveness of the two proposed algorithms by showing that they show better coverage compared to domain independent planners for numeric planning.

**Strengths:**

- The paper shows experimentally that their proposed learned heuristics shows better coverage, and that is significant.
- The approach is novel and paper is also novel (except for the recent related work which the authors cover in the intro)
- The authors do a decent job with explaining the literature and citing appropriately.
- The use of classical machine learning due to being cheap and interpretable

**Weaknesses:**

The paper presentation can be improved.  Examples below (no particular order):

1. The abstract states “… in comparison to domain-independent planners” whereas it is more informative to state domain-independent numeric planners. (Other places did indicate numeric planners).
2. The paper can be dense in various places (section 5 for example)
3. Sometimes the authors miss details that can be helpful in understanding the paper.  For example, what is M(3h||3n) can you explain? Maybe it would be good to give a bit of explanations on each of the configurations.
4. Fig 1, why x, y, z are not shown as blocks, the figure was confusing at first specially given that there are dotted block spaces above y and z with only 2 available blocks whereas the limit is 3 blocks.
5. In the Table2 caption, mention which ones are your proposed planners and the two variations (rank/cost).
6. minor typo: “Requires requires” repeated in section 3.1

**Questions:**

1. Can you claim theoretically on admissibility, safe pruning, or tractability of the proposed algorithms? I find it strange that the authors refer to prior work re theory. At least a proof sketch can be given which refers to prior work.
2. Related to 1, can you prove the complexity on the WL algorithm (right before section 4 starts).
3. Can you please explain with an example how one should read Table 1.
4. Which planner is h^LMCUT used in. h^LMCUT is a heuristics, not a planner, right? Same with the other heuristics mentioned in the figure, not sure if it is accurate to refer to them as planners.
5. In section 8, you mention in the last sentence, one can learn forms of domain knowledge different from heuristic functions, can you give an example.

**Limitations:**

The paper covers that.

---

> ### Author Rebuttal · Authors · 2024-08-03
>
> We thank the reviewer for the suggestions and questions for helping clarify the paper.
>
> ## Weaknesses
>
> We agree that the paper presentation can be improved and found it challenging to fit all of the material in the page limit. We will make good use of an extra page to address the presentation issues mentioned with more detail and illustrations.
>
> The dotted boxes in Fig. 1 represent the location of the other blocks if the optimal plan was followed.
>
> ## Questions
>
> Q1. Learning methods cannot guarantee any admissibility of heuristics. The learned heuristics are safe because predicted values are never infinite. The proposed learning algorithms are tractable because they make use of polynomial architectures. The only exception is the training criteria which may be intractable (e.g. solving a MIP optimally) but we can specify a timeout. The underlying search algorithms which we employ with our proposed learning algorithms are in the worst case intractable because (numeric) planning is in general intractable.
>
> Q2. The complexity of the original WL algorithm is given in the original WL paper and our paragraph before Sec. 4 extends this proof to show the complexity of our ccWL algorithm is the same as that of the original WL algorithm.
>
> Q3. The purpose of the table is to give an idea of how much larger / more difficult the testing problems are, in comparison with the training problems. As an example when reading the first row: The training problems for Blocksworld have between 2-11 blocks, while the testing problems have between 5 and 488 blocks. Furthermore, the optimal plan lengths (number of actions) for the training problems range from 2-34, while planners return between plans with lengths between 8-662.
>
> Q4. You are correct that a heuristic itself is not a planner. However, the “heuristics” in Table 2 are shorthand for the configurations of planners described in Sec. 6.2 as the notation is cumbersome if the planners were also mentioned. For example h^{mrp}+hj and M(3h||3n) are both different configurations of the ENHSP planner. As mentioned in lines 287-288, h^LMCUT is used in Numeric Fast Downward.
>
> We will make this clear in the figure and table captions.
>
> Q5. One simple example is to learn an action policy instead of a heuristic. We further list below several examples and related work in planning where we can learn domain knowledge in forms different from heuristic functions. All these methods can be used with our ccWL features (Sec. 3.2.) since they are agnostic to the downstream ML task.
>
> - policy rules [1], which are implication statements of the form (Condition($s$) -> Effect($s, s’$)) which tells us that an action $a$ should be applied in state $s$, leading to another state $s’$, if Condition($s$) and Effect($s, s’$) holds. Both the condition and effect can be some learned funcion over state features, such as those generated by the ccWL algorithm (Sec. 3.2)
> - policy sketches [2], a generation of policy rules by viewing the learned (Condition($s$) -> Effect($s, s’$)) statements as subgoals rather than direct action policies
> - portfolios [3] which learn what planner configurations work best for a given domain
> - task transformations such as learning to partially ground problems [4] or ignore objects [5]
>
> [1] Guillem Francès, Blai Bonet, Hector Geffner: Learning General Planning Policies from Small Examples Without Supervision. AAAI 2021: 11801-11808
>
> [2] Dominik Drexler, Jendrik Seipp, Hector Geffner: Learning Sketches for Decomposing Planning Problems into Subproblems of Bounded Width. ICAPS 2022: 62-70
>
> [3] Tengfei Ma, Patrick Ferber, Siyu Huo, Jie Chen, Michael Katz: Online Planner Selection with Graph Neural Networks and Adaptive Scheduling. AAAI 2020: 5077-5084
>
> [4] Daniel Gnad, Álvaro Torralba, Martín Ariel Domínguez, Carlos Areces, Facundo Bustos: Learning How to Ground a Plan - Partial Grounding in Classical Planning. AAAI 2019: 7602-7609
>
> [5] Tom Silver, Rohan Chitnis, Aidan Curtis, Joshua B. Tenenbaum, Tomás Lozano-Pérez, Leslie Pack Kaelbling: Planning with Learned Object Importance in Large Problem Instances using Graph Neural Networks. AAAI 2021: 11962-11971

---

> > ### Comment · Reviewer_R2Rg · 2024-08-08
> >
> > Thank you for answering my questions. I have no further questions.

---

### Official Review · Reviewer_YeVP · 2024-07-17

**Soundness:** 3
**Presentation:** 3
**Contribution:** 2
**Rating:** 4
**Confidence:** 2

**Summary:**

The authors introduce a method to generate features for planning tasks that involve numerical variables. These features can then be used with machine learning to learn a heuristic function from a set of training examples. Architectures used for learning include Gaussian processes and graph neural networks. The authors also introduce a method for learning to rank states and do a search based on the ranking instead of the cost-to-go. Results show that, for benchmarks modified to include numerical variables, learning a heuristic function with Gaussian process regression and ranking performs significantly better than planners that do not make use of a numerical representation.

**Strengths:**

Ranking states instead of learning cost-to-go has shown promise. The paper presents a novel ranking method that can be combined with machine learning. The ranking method was significantly better than the corresponding non-ranking method. This could have broader implications for machine learning applied to planning.

**Weaknesses:**

The only learning approach that performed better than the baseline planners was the Gaussian process regression with ranking, while Gaussian process regression with cost-to-go performed better than three out of four baseline planners. This makes it appear as if ranking is contributing to the overall success and not the numerical representation and learning.

**Questions:**

Do any of these baseline planners make use of ranking?

Is it possible that the main increase in performance is due to ranking and not the numerical representation and learning?

On line 152, V is comprised of G, while, on line 151, it says that G is comprised of V. Do these two Gs represent different concepts?

**Limitations:**

The learning approach relies on supervised learning, which assumes a planner exists that can already solve problems and may limit performance to what the existing planner can solve in a given time limit. On the other hand, research using deep reinforcement learning does not assume the existence of any solver.

---

> ### Author Rebuttal · Authors · 2024-08-03
>
> We thank the reviewer for the comments and questions for helping us identify where we could improve our paper’s clarity.
>
> ## Weaknesses clarifications
>
> > This makes it appear as if ranking is contributing to the overall success and not the numerical representation and learning.
>
> - We would first like to clarify that ranking is a learning concept. More specifically, we defined it as an optimisation problem using a Mixed Integer Program with *training data* with the aim of finding weights for the learning models. The simple analogy is that regression is a method for finding a function that best fits the training data, where one can define it as an optimisation problem with the MSE loss on some given training data.
>
> - Secondly, we note that Gaussian process regression (GPR) and ranking does not go together. Instead, the description of the models in the review should be changed as follows:
>
> > GPR with ranking
>
> should be changed to "ccWL algorithm (Sec. 3.2) with ranking"
> > GPR with cost-to-go
>
> should be changed to "ccWL algorithm with GPR representing cost-to-go"
>
> Please see the pdf of our global rebuttal for helping clarify the general pipelines of the models.
>
> - Lastly, we note that the strongest planner baselines (M(3h||3n), M-FF and h^{MRP+hj}) are representative of the state-of-the-art for numeric planning. They use additional tricks such as helpful and macro actions, multi-queue search, and building novelty heuristics. We decided to not combine our work with these techniques as they are orthogonal methods for improving performance which are also applicable to our methods. As mentioned on line 313, direct comparison of our results is only possible with h^add and h^RMP. All of our learning configurations clearly dominate them.
>
> We will make these details clearer in the additional page with extra explanations and illustrations.
>
> ## Questions
>
> Q1. The baseline planners are state of the art heuristic search planners, which do not make use of ranking or learning.
>
> Q2. Ranking is part of the learning process. Thus, without the numerical representation and learning, the models cannot employ ranking.
>
> Q3. Thanks for clarifying this conflict of notation. You are correct that the Gs represent different concepts, and we will fix this.
>
> The $G$ and $\mathcal{V}$ in line 151 refer to graph mentioned in line 124, while the $G$ and $V(g)$ in line 152 refer to the goals and the variables associated with the goal condition, where the $V(\cdot)$ notation is introduced in line 66. We will use different notations and make this clearer in the final version of the paper.
>
> ## Limitations. (Comparison with Reinforcement Learning)
>
> We first like to mention that although the paper focuses on supervised learning, our methods are applicable to RL settings as well. More specifically, the graphical representation of the task (assuming a factored state representation) and the GNN model, can be applied with typical RL methodologies such as PPO and DQN. We decided to focus on supervised learning over RL because generating optimal plans from the provided training tasks generally take a matter of seconds, with a few outliers taking more than a minute.
>
> Secondly, regarding pros and cons of RL, it has been shown in various works (e.g. the NeurIPS Flatland challenge [1], see RL vs non-RL winners, and a Deepmind paper on planning and RL methods for manipulation planning [2], where RL methods struggle to solve the simplest benchmark problems) that symbolic solvers outperform RL methods in scaling whenever a model is given.
>
>
> [1] NeurIPS 2020 Flatland Winners. https://discourse.aicrowd.com/t/neurips-2020-flatland-winners/4010
>
> [2] Ken Kansky, Skanda Vaidyanath, Scott Swingle, Xinghua Lou, Miguel Lázaro-Gredilla, Dileep George:
> PushWorld: A benchmark for manipulation planning with tools and movable obstacles. CoRR abs/2301.10289 (2023)

---

### Author Rebuttal · Authors · 2024-08-04

We thank all reviewers for their reviews and suggestions for improving our paper.

We noticed that the common weakness pointed out by reviewers is that our paper could make use of additional details or illustrations to better explain our methods. We agree and believe it was difficult to fit this in the page limit for the submission. We will make use of the extra page available for the final version to handle their suggestions.

Furthermore, we have attached a figure in the global PDF to visually summarise our proposed approaches and showing
1. how the separate sections and equations tie together into the final products, and
2. indicating where learning is involved.

---

### Decision · Program_Chairs · 2024-09-25

**Decision:**

Accept (poster)

**Comment:**

Overall, the technical contributions are solid and the evaluation shows strong results compared to state-of-the-art planners.

A common theme of the reviews was that the presentation is dense and can be improved. The authors have been receptive to this feedback and appear committed to improving the presentation.

There were also concerns about the relevance of the problem and the limited related work to compare to. These are definitely valid concerns when we see an uncommon problem being studied. In this case, the ACs assessment is that the problem of "symbolic planning for numeric planning problems" is a well-established area in a less populated corner of the AI world (e.g. the ICAPS community). It could be viewed as a positive step to bring this problem to the NeurIPS community. It is hard to predict the impact, but I can imagine a number of directions other researchers might advance the area.